# Telmisartan Inhibits Cell Proliferation and Tumor Growth of Esophageal Squamous Cell Carcinoma by Inducing S-Phase Arrest In Vitro and In Vivo

**DOI:** 10.3390/ijms20133197

**Published:** 2019-06-29

**Authors:** Takanori Matsui, Taiga Chiyo, Hideki Kobara, Shintaro Fujihara, Koji Fujita, Daisuke Namima, Mai Nakahara, Nobuya Kobayashi, Noriko Nishiyama, Tatsuo Yachida, Asahiro Morishita, Hisakazu Iwama, Tsutomu Masaki

**Affiliations:** 1Department of Gastroenterology and Neurology, Kagawa University, Kagawa 761-0793, Japan; 2Life Science Research Center, Kagawa University, Kagawa 761-0793, Japan

**Keywords:** esophageal squamous cell carcinoma, telmisartan, cyclin, angiotensin II type 1 receptor blocker, cell cycle arrest

## Abstract

Esophageal squamous cell carcinoma (ESCC) is the most common primary esophageal malignancy. Telmisartan, an angiotensin II type 1 (AT1) receptor blocker (ARB) and a widely used antihypertensive, has been shown to inhibit proliferation of various cancer types. This study evaluated the effects of telmisartan on human ESCC cell proliferation in vitro and in vivo and sought to identify the microRNAs (miRNAs) involved in these antitumor effects. We examined the effects of telmisartan on three human ESCC cell lines (KYSE150, KYSE180, and KYSE850). Telmisartan inhibited proliferation of these three cell lines by inducing S-phase arrest, which was accompanied by decreased expression of cyclin A2, cyclin-dependent kinase 2, and other cell cycle-related proteins. Additionally, telmisartan reduced levels of phosphorylated ErbB3 and thrombospondin-1 in KYSE180 cells. Furthermore, expression of miRNAs was remarkably altered by telmisartan in vitro. Telmisartan also inhibited tumor growth in vivo in a xenograft mouse model. In conclusion, telmisartan inhibited cell proliferation and tumor growth in ESCC cells by inducing cell-cycle arrest.

## 1. Introduction

Esophageal cancer is the seventh most common cancer worldwide; more than half a million new cases were diagnosed in 2018, and its mortality is the sixth highest of all cancers [1]. The mortality rate is high because esophageal cancer is usually diagnosed at a late stage [2]. Esophageal cancer is classified into two major types: Esophageal squamous cell carcinoma (ESCC) and esophageal adenocarcinoma (EAC). ESCC is the most common subtype and accounts for 70% of cases worldwide. The highest rates occur in a so-called “esophageal cancer belt” that stretches from Northern Iran through the Central Asian republics to North-Central China [3]. Despite advances in diagnosis and treatment, the five-year overall survival rate for persons with esophageal cancer is only 15–20% worldwide [4]. Many genetic abnormalities are associated with ESCC, as are altered expressions of the cell-cycle regulator cyclin and cyclin-dependent kinases (CDKs). Particularly, overexpressed cyclin-A may induce resistance to radiotherapy and poor prognosis in patients with ESCC [5]. Moreover, cyclin-A overexpression may decrease sensitivity to paclitaxel-based chemotherapy compared with ESCC patients with lower expression [6]. 

Several chemo-preventive agents are associated with decreased risk of esophageal cancer, including proton pump inhibitors, aspirin, nonsteroidal anti-inflammatory drugs (NSAIDs), and statins [7]. Angiotensin II receptor type 1 (AT1) receptor blocker (ARB) is widely used as an antihypertensive drug. Recent studies have shown that angiotensin II is associated with cancer progression, and ARB inhibits tumor growth by antagonizing the AT1 receptor [8,9,10]. Furthermore, ARBs reportedly improve overall survival in patients with ESCC who receive esophagectomies [11]. Telmisartan, which is an ARB, inhibited growth of various cancer cell types, both in vitro and in vivo, including lung cancer [12], endometrial cancer [13], urological cancer [14,15], hepatocellular carcinoma [16], and cholangiocarcinoma cells [17], and in our present study. It inhibited cancer proliferation via DNA-binding activity of PPAR-γ in lung cancer [8], induced apoptosis in urological and gynecologic cancer cell lines [13,15], and induced autophagy in adult T-cell leukemia [18]. A recent study revealed an association between telmisartan and inhibition of EAC cell proliferation via the cell cycle, growth factors, and microRNA (miRNA) regulation in vitro and in vivo [19], although the effect of telmisartan on ESCC remained unclear.

Here we present the result of the antitumor effects of telmisartan in human ESCC cell lines and xenograft models. We found that telmisartan induced cell-cycle arrest of ESCC cell lines but did not induce apoptosis. Telmisartan causes S-phase cell-cycle arrest by decreasing levels of cyclin A2, CDK2, and other cell-cycle-related proteins. We also examined the mechanisms of antitumor effects, including receptor tyrosine kinases (RTKs), angiogenesis, and miRNAs, as much as possible. 

## 2. Results

### 2.1. Telmisartan Inhibits the Proliferation Ability of Human ESCC Cells in Vitro

To investigate the effect of telmisartan in three human ESCC cell lines (KYSE150, KYSE180, KYSE850), cells were cultured in 2% FBS and treated with 0, 10, 50, or 100 µM of telmisartan for 48 h. Telmisartan dose-dependently reduced proliferation of the three ESCC human cell lines (Figure 1).

### 2.2. Telmisartan Induced Cell-Cycle Arrest in S Phase and Regulated Cell-Cycle-Related Proteins

To examine whether growth inhibition was due to cell-cycle change, we investigated the cell-cycle profiles of KYSE180 cells 24 h after treatment, with or without 50 µM telmisartan, using flow cytometry. Treatment with 50 µM telmisartan increased the percentage of cells in S phase and decreased dramatically the percentage of cells in G_2_/M phase at 24 h after treatment (Figure 2). 

The effects of telmisartan on expression of cell-cycle regulatory proteins were investigated by western blotting. KYSE180 cells were treated with or without 50 µM telmisartan for 24 h. Expressions of Cyclin A2 and CDK2 (key proteins in the S to G_2_ phase transition), and of Cyclin B1 and CDK1 (key proteins in the G_2_ to M phase transition) were significantly reduced in treated cells (Figure 3). These results suggest that telmisartan inhibits cell-cycle progression from S to G_2_/M phase by decreasing expression of Cyclin A2 and Cdk2 in human ESCC cells.

### 2.3. Telmisartan Does Not Promote KYSE180 Cell Apoptosis

To further investigate the anti-cancer effect of telmisartan on KYSE180 cells, we detected and quantified apoptotic cells after treatment with 50 µM telmisartan for 24 h, using flow cytometry (Figure 4). The percentage of apoptotic cells was not increased in treated KYSE180 cells compared with DMSO-treated controls. This result demonstrated that telmisartan did not induce apoptosis of KYSE180 cells.

### 2.4. Telmisartan Affects the p-ErbB3 Level in KYSE180 Cells

We performed a p-RTK array to identify key RTKs associated with the anti-cancer effects of telmisartan. We analyzed KYSE180 cells that were treated with 50 µM telmisartan, using the antibody array, which analyzed the expressions of 49 activated RTKs (Figure 5A). Telmisartan reduced the expression of p-ErbB3 in KYSE180 cells (Figure 5B). Therefore, telmisartan may reduce proteins related to the cell-cycle by inhibiting phosphorylation of ErbB3. Densitometry showed that p-ErbB3 intensity for telmisartan-treated KYSE 180 cells was 5% of that for untreated cells (Figure 5C). 

### 2.5. Telmisartan Affected the Thrombospondin-1 (TSP-1) Level in KYSE180 Cells

We performed an angiogenesis antibody array to identify key angiogenesis-related molecules associated with the anti-cancer effects of telmisartan. KYSE180 cells treated with 50 µM telmisartan were analyzed using the antibody array to screen expression of 56 angiogenesis-related proteins (Figure 6A). Telmisartan decreased the TSp-1 level in KYSE180 cells (Figure 6B). Densitometry showed that the intensity of the TSp-1 for the telmisartan-treated KYSE 180 cells was 36% of that for untreated cells (Figure 6C). 

### 2.6. Effect of Telmisartan on miRNA Expression

We analyzed expression levels of 2555 miRNAs in treated and untreated KYSE180 cells, using a custom microarray platform. Treatment with 50 µM telmisartan for 48 h upregulated expression of 180 miRNAs and downregulated expression of 184 miRNAs in KYSE180 cells. Unsupervised hierarchical clustering analysis was conducted using Pearson’s correlation; the result showed that the treated cells clustered separately from the control cells (Figure 7). We identified 36 miRNAs that were significantly upregulated and 23 miRNAs that were significantly downregulated in treated cells, compared with controls (Table 1). 

### 2.7. Telmisartan Inhibits Tumor Growth In Vivo

We subcutaneously injected nude mice with KYSE180 cells, followed by i.p injection of telmisartan at 50 or 10 µg/day, or DMSO control, to determine whether telmisartan inhibited tumor proliferation in vivo. The high-dose (50 µg/day) telmisartan treatment inhibited tumor growth; tumor volumes in the high-dose group were 74.2% lower than the tumor volumes of the control group (Figure 8A,B). The low-dose group tended to have smaller tumor volume than the control group, but not significantly so. All animals survived to the end of the experiment, and telmisartan did not affect body weight.

In immunohistochemistry examinations, percentages of PCNA+ cells were significantly less in the low-dose and high-dose groups than in the control group (Figure 8C,D).

## 3. Discussion

In the present study, we investigated the antitumor effect of telmisartan on ESCC growth in vitro and in vivo. A time- and dose-dependent anti-proliferation effect by telmisartan was seen in three ESCC cell lines, and a similar growth-inhibitory effect was seen in a telmisartan-treated ESCC xenograft model. The findings show that telmisartan induces S-phase arrest by decreasing expression of cell-cycle regulatory proteins in ESCC. These results are consistent with reported findings of the antitumor effect of telmisartan on EAC cell lines and tumor tissues [19]. More importantly, the antiproliferative effect of telmisartan is further validated for ESCC. 

To our knowledge, this is the first study to show that telmisartan inhibits ESCC cell proliferation and tumor growth in vitro and in vivo.

Telmisartan, an ARB, is commonly used all over the world as an antihypertensive agent. Recent investigations have shown that telmisartan inhibits cancer proliferation for several cancer types in vitro [12,14,15,20] and tumor growth in vivo [13,16,17,19], including our studies. Telmisartan has been shown to inhibit cell proliferation of OE19, OE33, and SK-GT4 on EAC [19], which is the predominant type of esophageal cancer in North America and Europe. In accordance with this finding, our results show the anti-tumor effect of telmisartan in the KYSE150, KYSE180, KYSE850 ESCC cell lines. In particular, as KYSE180 was the least sensitive of the three cell lines to time- and dose-dependent telmisartan-mediated cell death, we mainly chose KYSE180 as the model cell line in this examination.

Cell-cycle dysregulation and strong proliferation are important characteristics of malignant tumor cells, and cell-cycle arrest is a major anti-tumor mechanism. Cyclins are essential proteins in cell-cycle regulation, and cell proliferation is controlled by protein complexes composed of CDKs [21,22]. Cyclin A is a CDK that modulates the cell cycle and cell proliferation. S-phase progression depends on its interaction with cyclin A [23,24]. Telmisartan induced cell-cycle arrest at the S phase in the current study. Additionally, levels of the cell-cycle regulatory proteins cyclin A2 and CDK2 were substantially reduced. Furthermore, expression of PCNA, which is closely related to DNA synthesis in S phase, was decreased in the in vivo investigation. Several examinations have shown that telmisartan induces cell-cycle arrest in G_0_/G_1_ phase by reducing cyclin D1 [17,19,20,25]. Compared with our present study, this discrepancy may reflect differences of differentiation, or cell type in in vitro models. Thus, this finding suggests that these proteins—major cell-cycle regulators—could facilitate the telmisartan-mediated antitumor effect by inducing S-phase arrest in ESCC cells.

Previous studies have shown that telmisartan inhibits cell proliferation by inducing apoptosis in endometrial [13], ovarian [19], urological [14,15], lung [12,25], and colon [26] cancer cells; other studies have shown that telmisartan induces cell-cycle arrest, but not apoptosis, in EAC and cholangiocarcinoma [17,20]. In the present investigation, telmisartan did not increase the percentage of apoptotic cells in KYSE180 cells. Additionally, telmisartan did not significantly increase activated caspase-3 in KYSE180 cells. These data suggest that telmisartan mainly inhibits ESCC cell proliferation by inducing cell-cycle arrest but not apoptosis. 

Receptor tyrosine kinases (RTKs) occupy key hubs in cell signaling networks, and they are overexpressed in many cancers. ErbB3 is the member of the human epidermal receptor family, which plays essential roles in cell proliferation and survival. ErbB3 interacts prominently with the phosphoinositide-3-kinase (PI3K)/AKT pathway, and its overexpression has been reported in many primary cancers, including carcinomas of the stomach, colon, pancreas, oral cavity, breast, ovarian, prostate, and lung ([27]. In addition, *ErbB3* downregulation or knockdown induces cell-cycle arrest and apoptosis in colon cancer cell lines [28,29]. Telmisartan reportedly decreased ErbB3 in hepatocellular carcinoma [16]. Similarly, we showed that telmisartan decreased ErbB3 in ESCC cells by p-RTK array. These findings suggest that decreased ErbB3 is involved in the antitumor effect of telmisartan in ESCC cells.

Our angiogenesis antibody array results showed telmisartan to decrease expression of TSP-1, which is a multifunctional protein with anti-angiogenesis properties. Overexpression of TSP-1 has been associated with TMN stage and regional lymph node involvement in ESCC, and with poor survival for ESCC patients in one study [30]; whereas another study associated low TSP-1 expression with shorter progression-free survival after surgery in ESCC patients [31]. The role of TSP-1 is not consistent in other carcinomas, so its role in ESCC cells is also unclear. Decreased TSP-1 expression in telmisartan-treated ESCC cells may have an antitumor effect, but TSP-1 expression might only be decreased in ESCC cells that escape the antitumor effects of telmisartan. 

MiRNAs are small non-coding RNA molecules that regulate the development and progression of various cancers [32]. To find out which miRNAs are associated with the antitumor effects of telmisartan, we used miRNA expression assays. Their results showed that the expression of miRNA in telmisartan-treated ESCC cells were significantly different from that in control cells (36 upregulated and 23 downregulated). We used miRDB (http://mirdb.org/) and DIANA TOOLS (http://snf-515788.vm.okeanos.grnet.gr), a widely used database, to identify the direct target genes of these miRNAs, and found that cell-cycle-related proteins were the target genes of several miRNAs that were upregulated upon telmisartan treatment. For example, miR-765 reportedly targets the *CDK2* gene, and inhibits cell migration and tube formation in osteosarcoma cells [33], and cell proliferation, migration, and invasion in breast carcinoma cells [34]. Thus, our results suggest that changes in the expression of these miRNAs enable the antitumor effect of telmisartan.

In conclusion, our results indicate that telmisartan inhibits human ESCC cell proliferation and tumor growth by inducing S-phase cell-cycle arrest via regulation of cell-cycle-related proteins.

## 4. Material and Methods

### 4.1. Chemicals

Telmisartan was purchased from Tokyo Chemical Industry Co. (Tokyo, Japan). We prepared a stock solution of telmisartan by diluting it to 10 mM with dimethyl sulfoxide (DMSO). The stock solution was stored at −20 °C.

### 4.2. Cell Lines and Cell Culture

We used three human ESCC cell lines, KYSE150 (Lot.12112008), KYSE180 (Lot.8012008), KYSE850 (Lot.8252001), which were obtained from Japanese Collection of Research Bioresources Cell Bank (Osaka, Japan) on 28 May 2013. All cell lines were used within 3 passages. The cells were grown in Ham’s F12 (Fujifilm Wako Pure Chemical Corporation, Osaka, Japan)/RPMI-1640 (Gibco Invitrogen, Carlsbad, CA, USA) supplemented with penicillin-streptomycin (100 µg/mL) and 2% fetal bovine serum (FBS), at 37 °C in a humidified atmosphere containing 5% CO_2_. 

### 4.3. Cell Proliferation Assay

Cell proliferation assays were conducted using the Cell Counting Kit-8 (Dojindo Laboratories, Kumamoto, Japan) according to the manufacturer’s instructions. Each of three cell lines was seeded into a well of a 96-well plates (1.0 × 10^4^ cells/well) and was cultured in 100 µL of each culture medium. After 24 h, the seeded cells were treated by adding 0, 10, 50, or 100 µM telmisartan into the culture medium, and were cultured for an additional 48 h at 37 °C 5% CO_2_. The control group added with 1 µL of DMSO. At the indicated time points, medium in each well was exchanged for 100 µL of each culture medium containing CCK-8 reagent, and the cells were incubated for 2 h. Absorbance was measured at 450 nm using an automated microplate reader.

### 4.4. Flow Cytometry Analysis of The Cell Cycle and Apoptosis

We conducted a flow cytometric analysis using the Cycle Phase Determination kit (Cayman Chemical Company, Ann Arbor, MI, USA) to evaluate the mechanism of the growth inhibition by telmisartan. KYSE180 cells were digested by 0.25% trypsin and plated in 100-mm-diameter dishes at 1.0 × 10^6^ cells/dish. After incubating for 24 h without FBS, KYSE180 cells were treated with 50 µM telmisartan or DMSO (control) for another 24 h, then harvested, washed with phosphate-buffered saline (PBS), suspended in 500 µL of PBS plus 10 µL of RNase A (250 µg/mL) and 10 µL of propidium iodide (PI) stain (100 µg/mL), and incubated for 30 min. 

To evaluate the apoptosis rate of KYSE180 cells, we used flow cytometry and an Annexin V-FITC Early Apoptosis Detection Kit (Cell Signaling Technology, Boston, MA, USA). KYSE180 cells were plated in 100-mm-diameter dish at 1.0 × 10^6^ cells/dish and treated with 50 µM telmisartan or DMSO control for 24 h. After incubating for 24 h, KYSE180 cells were harvested, washed with PBS and suspended in 96 µL Annexin V. After adding 1 µL Annexin V-FITC, 12.5 µL PI, 140 µL Annexin V and 214 µL PBS, we analyzed apoptosis and necrotic cell death. Staining was performed according to manufacturer’s protocol. Flow cytometry was conducted with a Cytomics FC 500 flow cytometer (Beckman Coulter, Indianapolis, IN, USA) equipped with an argon laser (480 nm). Cell percentages were analyzed with Kaluza software (Beckmann Coulter).

### 4.5. Western Blot Analysis

After treatment with 50 µM telmisartan or DMSO control for 24 h, KYSE180 cells were washed twice in PBS and lysed in PRO-PREP complete protease inhibitor mixture (iNtRON Biotechnology, Seongnam, Korea). Protein concentrations were quantified using a NanoDrop 2000 spectrofluorometer (Thermo Fisher Scientific, Waltham, MA, USA) according to the manufacturer’s protocols. Protein aliquots (1–10 µg) were suspended in sample buffer and heated to 100 °C for 5 min. Protein lysates were separated on 10% Tris-glycine gradient gels via SDS-polyacrylamide gel electrophoresis and transferred to polyvinylidene difluoride membranes by the wet transfer method. After blocking with non-fat milk for 1 h, the membranes were incubated with primary antibodies overnight. After being washed by TBS-T, the membranes were incubated with HRP-conjugated secondary antibodies.

Antibodies against cyclin A1 (ab53699) and cyclin A2 (ab38) were purchased from Abcam (Cambridge, UK); against cyclin B1 (#4138) and CDK1 (#77055) from Cell Signaling Technology (Boston, MA, USA); against cyclin D1 (SP4) (#MA5-14512) and cyclin E (HE-12) (MS-870-P1) from Thermo Fisher Scientific; against CDK2 (sc-163), CDK4 (sc-749) and CDK6 (sc-177) from Santa Cruz Biotechnology (Santa Cruz, CA, USA); and anti-β-actin (A5441) from Sigma-Aldrich (St. Louis, MO, USA). Secondary antibodies included HRP-linked anti-mouse and anti-rabbit IgG (Cell Signaling Technology, USA). 

### 4.6. Antibody Array of Phosphorylated Receptor Tyrosine Kinase (p-RTK) and Angiogenetic Protein Profiles

We collected KYSE180/control and KYSE180/telmisartan-treated (50 µM) cells for human p-RTK and angiogenesis antibody arrays. We evaluated levels of 49 phosphorylated receptor tyrosine kinases and 56 angiogenesis-associated proteins by using the human p-RTK and angiogenesis antibody array kits (R&D Systems, Minneapolis, MN, USA) according to manufacturer’s protocols. Densitometric ratio to the reference spot was calculated for each spot, and the comparison of control group and treatment group was performed using the densitometric ratio.

### 4.7. Analysis of the miRNA Array

After treatment with 50 µM telmisartan or DMSO control in serum-free medium for 24 h, total RNA was extracted from KYSE180 cells using a miRNeasy Mini Kit (Qiagen, Hilden, Germany) according to manufacturer’s protocols. 

RNA samples typically showed A_260/280_ ratios between 1.9 and 2.1, using an Agilent 2100 Bioanalyzer (Agilent Technologies, Santa Clara, CA, USA). After measuring RNA quantities using the RNA 6000 Nano kit (Agilent Technologies), samples were labeled using a miRCURY Hy3/Hy5 Power Labeling Kit (Exiqon A/S, Vedbaek, Denmark) and subsequently hybridized to a human miRNA Oligo chip (v.21; Toray Industries, Inc., Tokyo, Japan). The chips were scanned with the 3D-Gene Scanner 3000 (Toray Industries). We used 3D-Gene Extraction, version 1.2 software (Toray Industries) to read the raw intensity of the image. Changes in miRNA expression between telmisartan-treated and DMSO-control samples were analyzed with GeneSpringGX v.10.0 (Agilent Technologies). Quantile normalization was performed on the raw data that were greater than the background level. Differentially expressed miRNAs were determined by the Mann–Whitney U test. The false discovery rate was computed with the Benjamini–Hochberg procedure for multiple testing. Hierarchical clustering was performed using the furthest neighbor method, with the absolute uncentered Pearson’s correlation coefficient as a metric. We produced a heat map with the relative expression intensity for each miRNA, in which the base-2 logarithm of the intensity was median-centered for each row.

### 4.8. Xenograft Model Analysis

Animal experiments were performed according to the guidelines of the Committee on Experimental Animals of Kagawa University, (Kagawa, Japan). Female athymic mice (BALB/c-*nu/nu*; 6 weeks old, 20–25 g) were purchased from Japan SLC, Inc. (Shizuoka, Japan). The mice were bred under specific pathogen-free conditions using a laminar airflow rack. The mice had continuous free access to sterilized food (gamma-ray-irradiated food, CL-2; CLEA Japan, Inc. Tokyo, Japan) and autoclaved water. The mice were subcutaneously inoculated with KYSE180 cells (3.0 × 10^6^ cells/animal) in their flanks. Twelve days later, the xenografts were identifiable as masses with maximal diameters >4 mm. After measuring mass, the mice were randomly assigned to three groups. The telmisartan-treated group were intraperitoneally (i.p.) injected every day with 0.5 mg/kg (10 µg/day) or 2.5 mg/kg (50 µg/day) telmisartan for three weeks. The control group was i.p. injected with DMSO only for three weeks. Tumors were measured every three days. Tumor volume was calculated as the tumor length (mm) × tumor width (mm)^2^/2 [35]. All mice were sacrificed on Day 18 after treatment; all mice survived until then. 

### 4.9. Immunohistochemistry

Immunohistochemistry was performed on tumor tissues obtained from the xenografted mice. All tumors were fixed in 10% formaldehyde and embedded in paraffin before histological sectioning. The samples were then cut into 4-µm sections and deparaffinized, rehydrated, and subjected to immunohistochemistry studies. Tissue sections were incubated with primary antibodies (proliferating cell nuclear antigen [PCNA], Vector Laboratories, Burlingame, CA, USA). Sections were washed and incubated with secondary antibody (Vector Laboratories) and then with streptavidin-peroxidase solution. Color reactions involved the use of 3,3′-diaminobenzidine (DAB) with Mayer’s hematoxylin counterstaining. The specificity of immunostaining was evaluated using non-immune mouse IgG as a negative control for the primary antibody. Images were captured using an Olympus BX51 microscope and Olympus DP72 camera (Olympus, Tokyo, Japan). Nuclear labeling index for the PCNA+ cells (positive nuclei/total counted) was determined by evaluating at least 500 hepatocytes at random in the microscopic field by 2 observers (Matsui T and Masaki T).

### 4.10. Statistical Analyses

All statistical analyses were performed using GraphPad Prism 7.0 (GraphPad Software, Inc, LA Jolla, CA, USA). Comparisons between the treatment groups and control groups were performed by two-tailed paired or unpaired Student’s *t*-test or two-way ANOVA. *P* < 0.05 was considered significant.

## Figures and Tables

**Figure 1 ijms-20-03197-f001:**
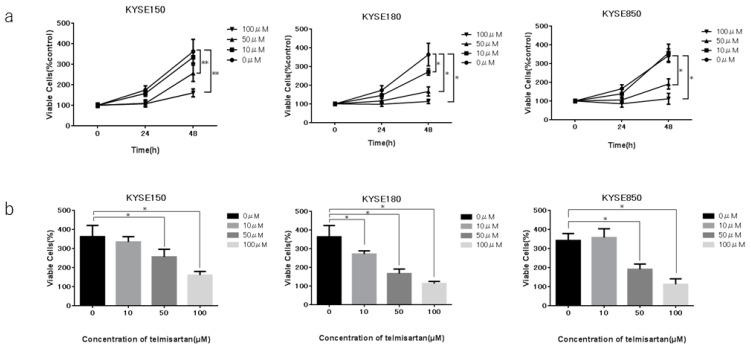
Effects of telmisartan on proliferation of ESCC cell lines in vitro. Telmisartan inhibited proliferation of ESCC cells. Viability of treated cells differed significantly from that of control cells. (**a**) KYSE150, KYSE180, and KYSE850 cells were seeded in 96-well plate (1.0 × 10^4^ cells/well). After 24 h, telmisartan (10, 50, or 100 µM) or DMSO were added to the fresh culture medium. Cell viability was assayed daily from 0 to 48 h. (**b**) Cell viability of ESCC cells at 48 h. (**p* < 0.01).

**Figure 2 ijms-20-03197-f002:**
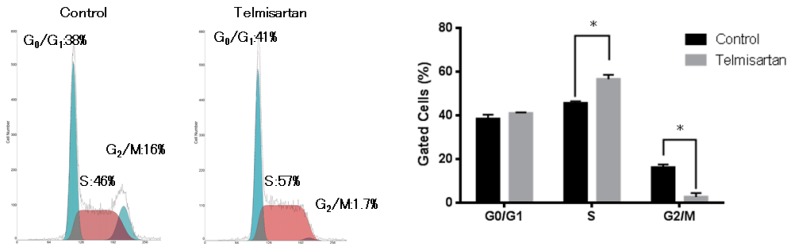
Flow cytometric analysis of KYSE180 cells treated with 50 µM telmisartan at 24 h. Telmisartan increased the population of cells in the S phase and decreased the population of cells in the G_2_/M phase. Telmisartan blocks cell-cycle progression to G_2_/M from S phase. (**p* < 0.01).

**Figure 3 ijms-20-03197-f003:**
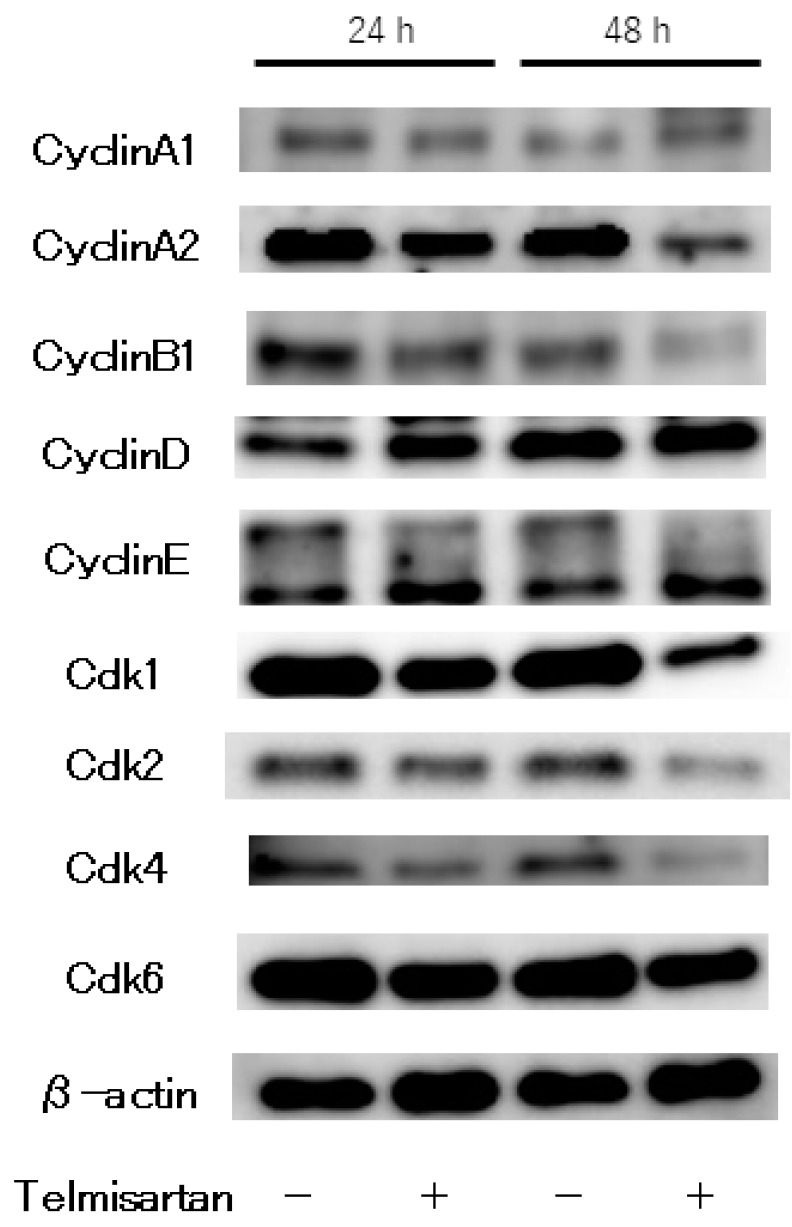
Western blot analysis of cell-cycle regulatory proteins in KYSE180 cells treated with 50 µM telmisartan. Expression levels of Cyclin A2, Cyclin B1, CDK1, CDK2, CDK4 were decreased in treated cells.

**Figure 4 ijms-20-03197-f004:**
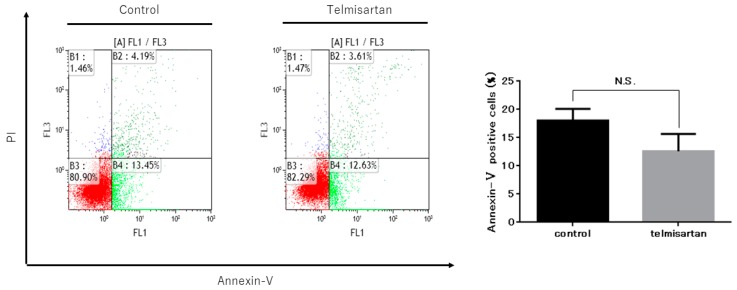
Telmisartan does not promote apoptosis in KYSE180 cells. Flow cytometry assessment of apoptosis of KYSE180 cells treated with 50 µM telmisartan at 24 h. Percentages of Annexin V+ cells did not significantly differ between control cells and telmisartan-treated cells. Apoptosis in KYSE180 is not induced by telmisartan.

**Figure 5 ijms-20-03197-f005:**
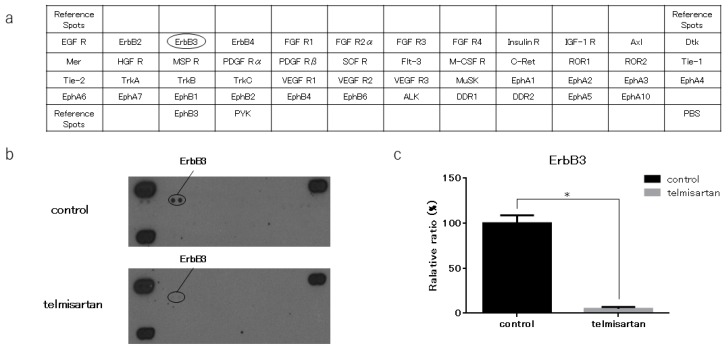
Result of p-RTK array for KYSE180 cells. (**a**) Template shows locations of tyrosine kinase antibodies on human p-RTK array. (**b**) p-ErbB3 expression was decreased in KYSE180 cells treated with 50 µM telmisartan at 24 h. (**c**) Densitometric ratio of telmisartan-treated group to non-treated group for p-ErbB3 spots. * *p* < 0.01.

**Figure 6 ijms-20-03197-f006:**
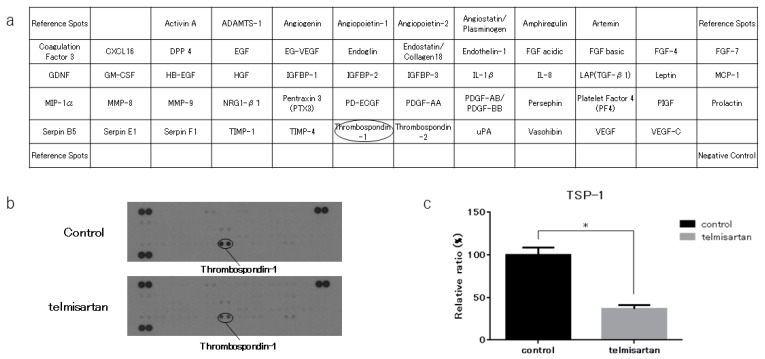
Angiogenesis antibody array in KYSE180 cells. (**a**) Template shows locations of angiogenesis antibodies on human angiogenesis array. (**b**) TSP-1 expression was decreased in KYSE180 cells treated with 50 µM telmisartan at 24 h. (**c**) Densitometric ratio of telmisartan-treated group to the non-treated group for TSP-1 spots. * *p* < 0.01.

**Figure 7 ijms-20-03197-f007:**
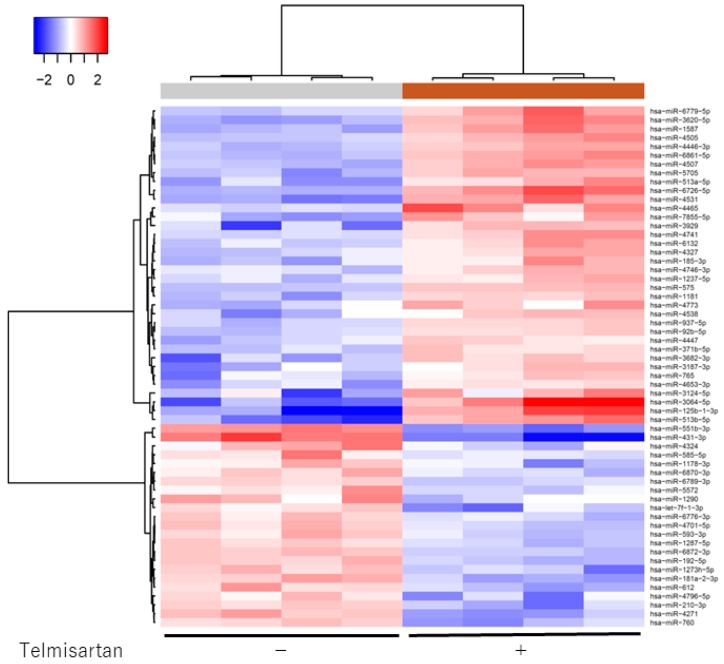
Hierarchical clustering of KYSE180 cells treated with or without telmisartan. KYSE180 cells were clustered according to expression profiles of 60 miRNAs differentially expressed by KYSE180 cells, with or without telmisartan. Top: miRNA clustering color scale indicates relative miRNA expression levels; red: high expression, blue: low expression.

**Figure 8 ijms-20-03197-f008:**
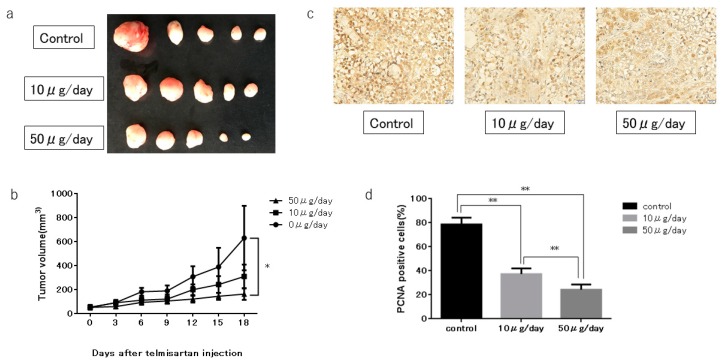
KYSE180 tumors in nude mice treated with telmisartan. (**a**) Representative images of gross KYSE180 tumors treated with telmisartan (10 µg/day or 50 µg/day), or the control. (**b**) Growth curves of xenografts showing mean tumor volume of control and telmisartan at 10 µg/day or 50 µg/day). Tumors were significantly smaller in the high-dose (50 µg/day) group than in the control group. (**c**) Immunohistochemistry examination using PCNA staining (400× magnification). (**d**) Quantitative comparison of PCNA+ cells of the control, low-dose (10 µg/day) and high-dose (50 µg/day) groups. PCNA+ cells and total numbers of cells were counted from 5 fields (400× magnification). Percentages of PCNA+ cells were significantly smaller in low-dose and high-dose telmisartan treated groups than in the control group. * *p* < 0.05; ** *p* < 0.01.

**Table 1 ijms-20-03197-t001:** Statistical results and chromosomal location of miRNAs in KYSE180 cells treated with telmisartan or untreated.

miRNA	Fold Change (Treated/Nontreated)	*P*	Chromosome Site	FDR
**Upregulated**				
hsa-miR-3064-5p	10.83	0.0010282	17q23.3	0.011157148
hsa-miR-125b-1-3p	9.11	0.001271658	11q24.1	0.0125323
hsa-miR-513b-5p	6.09	0.000497916	Xq27.3	0.006177046
hsa-miR-4531	5.05	5.85714E-05	19q13.31	0.002005486
hsa-miR-6726-5p	5.03	9.25009E-05	1p36.33	0.002316916
hsa-miR-3620-5p	4.40	6.52537E-05	1q42.13	0.002053966
hsa-miR-1587	3.94	0.000113112	Xp11.4	0.002316916
hsa-miR-4465	3.48	0.002098485	6q24.1	0.016612376
hsa-miR-6861-5p	3.38	3.99841E-05	12q24.13	0.001651115
hsa-miR-6779-5p	3.27	0.000766206	17q12	0.009109338
hsa-miR-4507	3.26	3.66678E-05	14q32.33	0.001651115
hsa-miR-513a-5p	3.25	0.001965543	Xq27.3	0.016207521
hsa-miR-5705	3.05	0.000161489	4q22.1	0.002879892
hsa-miR-3929	3.02	0.007579917	18q12.2	0.037505255
hsa-miR-7855-5p	3.01	0.002769579	14q23.3	0.020416622
hsa-miR-3124-5p	2.94	0.030352961	1q44	0.093125929
hsa-miR-4505	2.92	0.000284337	14q24.3	0.004425312
hsa-miR-4446-3p	2.89	3.70271E-05	3q13.2	0.001651115
hsa-miR-3682-3p	2.80	0.003888419	2p16.2	0.024658098
hsa-miR-575	2.63	2.92538E-07	4q21.22	0.000220522
hsa-miR-185-3p	2.62	0.008315209	22q11.21	0.039764351
hsa-miR-4741	2.54	0.00091106	18q11.2	0.01012815
hsa-miR-6132	2.49	0.002121681	7q31.2	0.016612376
hsa-miR-1181	2.48	0.000436849	19p13.2	0.005665799
hsa-miR-4773	2.42	0.012530723	2q23.3	0.051415576
hsa-miR-3187-3p	2.39	0.009570854	19p13.3	0.043622441
hsa-miR-4327	2.33	0.00208623	21q22.11	0.016612376
hsa-miR-4538	2.21	0.016769506	14q32.33	0.062959199
hsa-miR-765	2.15	0.01548123	1q23.1	0.059425707
hsa-miR-937-5p	2.11	6.85217E-05	8q24.3	0.002053966
hsa-miR-4653-3p	2.10	0.006521157	7q22.1	0.034671494
hsa-miR-4746-3p	2.08	0.001577889	19p13.3	0.01392446
hsa-miR-371b-5p	2.07	0.000340744	19q13.42	0.004781581
hsa-miR-92b-5p	2.06	0.000206021	1q22	0.003391425
hsa-miR-1237-5p	2.05	0.004806337	11q13.1	0.027987923
hsa-miR-4447	2.03	0.001493554	3q13.31	0.013638253
**Downregulated**				
hsa-miR-431-3p	0.09	7.05876E-05	14q32.2	0.002053966
hsa-miR-551b-3p	0.18	3.55933E-06	3q26.2	0.000917897
hsa-miR-4271	0.30	0.000111417	3p21.31	0.002316916
hsa-miR-181a-2-3p	0.32	0.000142642	9q33.3	0.002713359
hsa-miR-612	0.38	0.00102969	11q13.1	0.011157148
hsa-miR-1273h-5p	0.39	0.002814441	16p12.1	0.020416622
hsa-miR-760	0.39	0.000827908	1p22.1	0.009576883
hsa-miR-192-5p	0.39	3.52356E-05	11q13.1	0.001651115
hsa-miR-210-3p	0.40	0.001969138	11p15.5	0.016207521
hsa-miR-6872-3p	0.42	1.83967E-05	3p21.31	0.00136536
hsa-let-7f-1-3p	0.42	0.008961426	9q22.32	0.041607982
hsa-miR-1290	0.43	0.010000273	1p36.13	0.044817978
hsa-miR-4324	0.43	0.029330891	19q13.33	0.090639866
hsa-miR-4701-5p	0.44	0.000306936	12q13.12	0.004529948
hsa-miR-4796-5p	0.45	0.015071207	3q13.31	0.058375353
hsa-miR-593-3p	0.45	0.001221347	7q32.1	0.012446109
hsa-miR-1287-5p	0.47	0.000148136	10q24.2	0.002756624
hsa-miR-6776-3p	0.48	0.001494273	17p13.3	0.013638253
hsa-miR-6870-3p	0.48	0.003873881	20p12.2	0.024658098
hsa-miR-1178-3p	0.49	0.021002746	12q24.23	0.072979103
hsa-miR-585-5p	0.49	0.021434077	5q35.1	0.073606371
hsa-miR-6789-3p	0.49	1.91406E-05	19p13.3	0.00136536
hsa-miR-5572	0.50	0.013211437	15q25.1	0.05284575

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
