# Peer review of "Telmisartan Inhibits Cell Proliferation and Tumor Growth of Esophageal Squamous Cell Carcinoma by Inducing S-Phase Arrest In Vitro and In Vivo"

_ijms, 2019, doi:10.3390/ijms20133197_

Round 1
Reviewer 1 Report
Use other more data bases, e.g.,
http://snf-515788.vm.okeanos.grnet.gr
to evaluate obtained miRNAs,
It will add more values to your manuscript.
Author Response
Reviewer 1
Point 1: Use other more data bases, e.g.,http://snf-515788.vm.okeanos.grnet.gr
to evaluate obtained miRNAs, it will add more values to your manuscript.
Response 1: Thank you very much for reviewing and your comments. We also evaluated miRNAs in DIANA TOOLS, we added that we used the database in the discussion paragraph (page 12, line 343-346).
Reviewer 2 Report
In this manuscript, Matsui et al., conducted a series of experiments to study the effect of Telmisartan on esophageal squamous cell carcinoma. They found that Telmisartan inhibits cell proliferation and tumor growth in a dose-dependent manner. While the results are indeed impressive, the manuscript should include the following information:
1. Please include information on authentication of cell lines
2. Please provide information on controls for the cell proliferation assay
3. Please include the details about antibodies used in Western Blot experiments
4. Fig 5 should consist of the expression of activated RTKs other than ErbB3 that would serve as a control for this experiment
5. Legend for Fig 5C is missing
Author Response
Reviewer 2
Point 1: Please include information on authentication of cell lines
Response: Thanks you very much for your advice. According your comments we added cell line lot number and purchase date (page 2, line 69-71).
Point 2: Please provide information on controls for the cell proliferation assay
Response: The control group added with 1µl of DMSO. According to your comments we revised our manuscript and made possible correction as well in method section (page2, line 80-81).
Point 3: Please include the details about antibodies used in Western Blot experiments
Response: We added the product number of antibodies used in Western Blot analysis, in method section (page 3, line 112-118).
Point 4: Fig 5 should consist of the expression of activated RTKs other than ErbB3 that would serve as a control for this experiment
Response: We agree with your indication. We attempt to evaluate each expression of activated RTKs by using image analysis tools, but each expression of them was not significantly observed except for ErbB3. The comparison was made using the densitometric ratio with reference spot. According to your comments we revised our manuscript and made possible correction in the method section (page 3, line124-126).
Point 5: Legend for Fig 5C is missing
Response: Thanks you very much for your advice. According to your comments we revised our entire manuscript and Figure for possible errors, and we made possible correction as well (page 8, line 229-231).